# Transformer Networks Enable Robust Generalization of Source Localization for EEG Measurements

## Abstract

An electroencephalogram (EEG) is an electrical measurement of brain activity using electrodes placed on the scalp surface. After EEG measurements are collected, numerical methods and algorithms can be employed to analyze these measurements and attempt to identify the source locations of brain activity. These traditional techniques often fail for measured data that are prone to noise. Recent techniques have employed neural network models to solve the localization problem for various use cases and data setups. These approaches, however, make underlying assumptions that make it difficult generalize the results past their original training setups. In this work, we present a transformer-based model for single- and multi-source localization that is specifically designed to deal with difficulties that arise in EEG data. Hundreds of thousands of simulated EEG measurement data are generated from known brain locations to train this machine learning model. We establish a training and evaluation framework for analyzing the effectiveness of the transformer model by explicitly considering the source region density, noise levels, drop out of electrodes, and other factors. Across these vast scenarios, the localization error of the transformer model is consistently lower than the other classical and machine learning approaches. Additionally, we perform a thorough ablation study on the network configuration and training pipeline. The code and data used in this work will be made publicly available upon publication.

## 1 Introduction

Arising from cognitive processes, the brain generates electricity when neurons are firing. When a large quantity of neurons are firing simultaneously, this can be measured by Electroencephalogram (EEG). EEG is a common technique in neuroscientific research to measure brain activity using electrodes placed on the scalp surface. EEG is sparsely sampled at different scalp locations to remotely monitor brain activity as potential difference measurements. Solving the location of the source given these sparse EEG measurements is an ill-posed inverse problem (Helmholtz, 1853).

Traditional approaches have used numerical methods and algorithms to analyze EEG measurements and identify the source locations of brain activity. These traditional techniques, however, can be very slow and often fail for measured data that are prone to noise and multi-source activations (Dannhauer et al., 2013). Recent approaches have utilized advances in machine learning and artificial neural networks to predict source locations by training on large amounts simulated data. Architectures involving convolutional layers (Hecker et al., 2021), fully-connected layers (Hecker et al., 2022), graph layers (Jiao et al., 2022), and have been explored.

Transformer models have been shown to provide state-of-the-art results in fields such as natural language processing and computer vision, but as of yet, the transformer architecture has been under explored for the source localization task. Transformer architectures that do exists (Zheng & Guan, 2023) follow the same fixed input and output size setup as other networks. This under-utilizes the unique features of the transformer that allow it to generalize well across datasets. For the input, the attention mechanism can operate on any number of tokens and in any order, even at runtime. For the output, the number of learned embeddings can explicitly determine the number of desired classifications. Such capabilities have direct application to real scenarios for collecting and processing

EEG data, where electrodes could be faulty or misconfigured and where multi-source accuracy is vital.

In this work, we present a novel transformer model for solving the inverse problem that is designed with such consideration in mind. Our transformer model represents each electrode as its own input token, and uses a number of learned embeddings to match the desired predicted source locations. We demonstrate the effectiveness of the transformer model in solving this inverse EEG problem for both single- and multi-source setups. We also show its robustness to the EEG electrode placement, noise in the EEG signal, and dropout of individual electrodes. At high noise levels, the transformer model can outperform all other methods in single-source localization by over 3mm in average MLE and by 12mm in multi-source localization.

To train the transformer, we create EEG simulations utilizing a head model of an individual and add varying levels of measurement noise. These EEG simulations require discretizing the brain into specific regions. Previous works have relied on representing the brain regions through simple grouping techniques such as voxelization (Hecker et al., 2021) or as divisions of icosohedral meshes. Such setups do not represent the complex folded surface of the brain or do not allow for direct control of the desired localization accuracy. To combat these issues, in this work, we use a Poisson-disc-like approach where desired spacing between regions can be approximately enforced. This allows for more direct control of the source region clustering. We used both a 10mm and 5mm spacing when evaluating the performance of the model.

Our transformer method is heavily evaluated, and the evaluations are designed to thoroughly understand the models abilities in difficult scenarios that appear in EEG data. The performance is measured for single- and multi-source tasks at high and low spacing between sources, high and low noise levels, and with possible electrode drop out. The method is compared to other techniques and consistently outperforms them in each unique scenario. This showcases the transformer model as a viable solution for EEG inverse problem and step towards generalization to real data.

In addition to the presented method, many variations are possible on the network configuration and training pipeline. This work explores many possible ablations to the method, including an iterative training approach, an unconstrained location prediction approach, and multiple alterations to the network configuration. We empirically found the that the presented approach was the best possible design and training setup, and the details of these alternate configurations are presented in the appendix.

In summary, our contributions are as follows:

- We present a unique transformer-based neural network that can handle variable number of inputs and explicitly define the number of source outputs.
- We present a novel approach for directly controlling the spacing between source regions on the brain for the forward model.
- We conduct through evaluations and comparisons, showing our model outperforms other architectures and methods on single- and multi-source prediction for high-noise and drop out scenarios.
- We provide a thorough ablation study of the network architecture and its effect on the supervised learning task.

## 2 RELATED WORKS AND BACKGROUND

### 2.1 EEG SOURCE LOCALIZATION

Over the past few decades, significant advances have been made in reconstructing neuronal sources from EEG data. These techniques rely on a forward model describing the physical and electromagnetic properties of head tissues and sensors. Simple spherical models provide an analytical approximation, but finite element modeling (FEM) offers higher numerical accuracy by capturing complex geometry and resistivity. The forward model underlies the inverse problem, where neuronal sources are estimated from EEG signals. In FEM head models, a dipolar current source simulates volume currents that reach the electrodes (Eichelbaum et al., 2014). Current dipoles, which approximate point-like flows, are widely used due to their flexibility in capturing diverse neuronal configurations.

Source configurations vary in strength and time, with peaks in activity marking concentrated brain processes. These may appear as a single source, bilateral sources, or multiple correlated sources. Different source localization methods address the inverse problem by uncovering the contribution of underlying sources at specific times.

Because the inverse problem is ill-posed, algorithms impose assumptions about number, location, and strength of sources. Numerically, whether the problem is under- or over-determined depends on the number of sources relative to independent measurements. In the over-determined case, a few discrete sources are estimated (e.g., dipole fit, beamformer), a method fairly noise-resistant but prone to missing or misattributing activity. In the under-determined case, a large number of dipoles—often oriented perpendicular to the cortical surface—are distributed across the brain to better explain EEG data. Here, minimal energy constraints (e.g., minimum norm estimate, MNE) reduce non-uniqueness, forming a distributed source model (Dannhauer et al., 2013). Additional temporal or statistical constraints further limit the solution space and improve robustness to noise.

A classical distributed method is eLORETA (Pascual-Marqui, 2007), known for robustness and practical utility (Jatoi et al., 2014). Like sLORETA (Pascual-Marqui et al., 2002), it minimizes L2-norms between predicted and measured EEG while regularizing source strength via Tikhonov methods. Although these approaches suffer from depth bias, weighting schemes and iterative matrices (e.g., in eLORETA) mitigate this issue. Distributed models remain noise-sensitive and often require averaging or preprocessing, though nonlinear and spatio-temporal strategies have been developed to address closely spaced or correlated sources.

We tested both distributed source reconstruction methods sLORETA as well as eLORETA on simulated data and found identical results, so we report eLORETA henceforth. We included also a discrete source reconstruction method called linearly constraint minimum variance (LCMV) beamformer (Van Veen et al., 1997). A more comprehensive review of classical techniques is provided by (Zorzos et al., 2021).

## 2.2 MACHINE LEARNING

Using machine learning to solve the inverse problem is a recent field of exploration. ConvDip (Hecker et al., 2021) presented the first convolutional neural network to solve EEG source localization. In their distribution-based spatial predictor, the brain is separated into voxels and the energy in each voxel for a single moment in time is predicted using the neural network. Many approaches would soon follow that would expand to different network architectures and types of prediction. (Hecker et al., 2022) presented a recurrent neural network with fully-connected layers to make spatial-temporal predictions. Later, a similar spatio-temporal predictor was designed using graph neural networks (Jiao et al., 2022). Other approaches included learning linear combinations of basis functions (Wei et al., 2021), using mass models (Sun et al., 2022), augmenting CNNs with finite element analysis (Delatolas et al., 2023), and modifying LSTMs (Sarah et al., 2023) or autoencoders (Huang et al., 2024; Liang et al., 2023). A summary of supervised learning approaches for electrical source imaging is presented in (Reynaud et al., 2024).

While many works have focused on the source localization task using CNNs and LSTMs, none have used Transformers for the spatial or temporal components. Transformers have been used heavily for EEG classification tasks with network architectures such as EEGFormer (Wan et al., 2023), EEG-ViT (Yang & Modesitt, 2023), EEG-Conformer (Song et al., 2023), and EEG-Deformer (Ding et al., 2025), but such successes have not yet translated to the source localization task. To our knowledge, only one other work has presented a transformer-based model for solving the EEG source localization task (Zheng & Guan, 2023). Their network, however, relies on the fixed input and distribution-based setups of prior works and doesn't provide any ablations on network configuration. By comparison, our transformer uses a learnable embedding tokens to separate single-source from multi-source localization, incorporates location into the input vectors, and is easily extendable to unconstrained location prediction. Additionally in this work, unique aspects of the transformer are explored to analyze aspects such as noise level effects on training and unique network configurations.

## 3 METHODOLOGY

For the localization of sources generating EEG measurements, the source location in the brain is only be determined by measured electrical potentials at the scalp surface. These sources cause electric current flow through the head tissues which manifests in the EEG potentials and can be modeled using a finite element model of the head tissue properties. The shape and extent of the head tissue properties can be derived from magnetic resonance imaging. In this work, we use the so called Ernie head model (Thielscher et al., 2015) to generate EEG data for source localization using classical (i.e., eLORETA) as well as machine learning approaches. There are specific considerations in configuring a machine learning model to solve the inverse problem. We explore those considerations, as well as detail our transformer model architecture, in the following subsections.

### 3.1 REGION CLUSTERING AND FORWARD SIMULATION

While a source signal could be generated continuously along any part of the brain surface, effective machine learning models need discretized outputs to learn and compute metrics on. Previous techniques have voxelized the brain into a 3-dimensional grid (Hecker et al., 2021) or use icosahedral divions, but this ignores the complex folded nature of the brain surface. For this work, we generate region clusters by distributing points equidistantly across the brain surface for a specific geodesic distance threshold (either 5mm or 10mm) in-between them. To determine each cluster center, we used the brain surface mesh nodes and their computed geodesic distances (using Dijkstra's algorithm) to remove any nodes from the calculations that fall within a geodesic distance threshold. We repeated this Poisson-disc-like approach for all remaining nodes. At the 10mm spacing, this results in 1803 regions represented by one dipole pointing outwards and perpendicular to the brain surface. We also repeated this procedure for a finer discretization with a 5mm spacing which resulted in 6790 regions as shown in Figure 1.

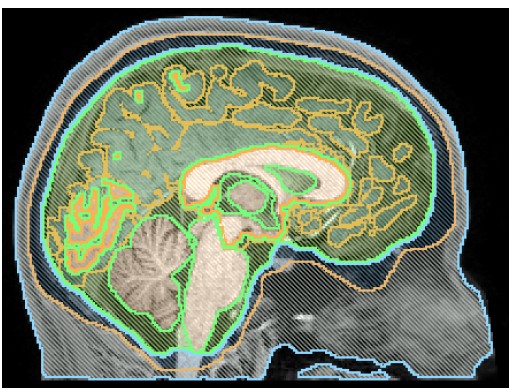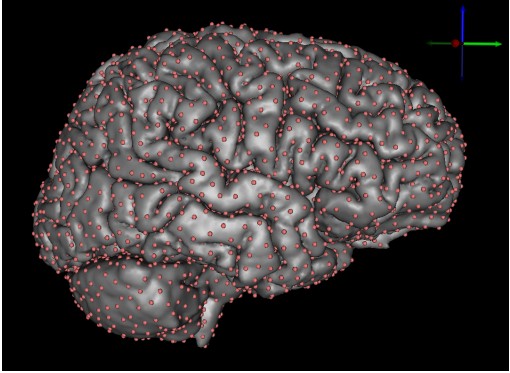

Figure 1: The modeled Ernie head is made up by many intricate tissues (right) and folded brain regions (left) as smooth surface as tetrahedral volume element discretiized from tissue-label voxels (left). We distributed source locations (shown as red dots) across the brain surfaces equally within 10mm (shown here) or 5mm geodesic distance between them using an in-hourse Possion disc-like approach.

We used the Ernie head model (part of SimNIBS software (Thielscher et al., 2015)), which comes with electrode scalp locations defined according to the 10-10 electrode placement system. The Ernie head model consists of 662 thousands nodes and 3.7 million tetrahedral elements. With standard tissue conductivities (Thielscher et al., 2015), we used the EIDORS-3D (Polydorides & Lionheart, 2002) package to compute a finite element stiffness matrix $\mathbf{A}$ and an in-house software (Gomez et al., 2021) to compute the right-hand side b. We then solved the equation system $\mathbf{A} \cdot \mathbf{x} = \mathbf{b}$ for $\mathbf{x}$ using the software SCIRun 4.7 (Parker & Johnson, 1995). The potential values coincide with the 76 electrode locations on the scalp and are stored as a column of the so called leadfield matrix for each dipole. Leadfield matrices were created, one for the 10mm (dimensions: 76×1803) and one for the 5mm source space (76×6790). Both leadfield matrices were used directly for our MATLAB-implemented classical (for eLORETA, LCMV) as well as for machine learning approaches. The regularization parameter for eLORETA and LCMV were logarithmically spaced between $10^{-5}$ and $10^{15}$ (MATLAB

command: $\alpha$=logspace(-5, 15, 10)). For eLORETA, we used Tikhonov regularization of the minimum norm estimates (i.e., equals to 10 regularization parameter denoted as $\alpha$; Pascual-Marqui (2007)) and for LCMV we additionally regularized the noise covariance matrix (i.e., $\alpha = \beta$ being the Tikhonov regularization factor of the covariance matrix; this equals to 10x10=100 different $(\alpha, \beta)$ parameter pair combinations).

## 3.2 Synthetic Noise, Training Setup, and Dropout

As is common for the inverse problem, we add synthetic measurement noise, relative the the EEG signal strength, to the simulated EEG measurement values. Rather than simply randomizing the strength, we train our network at different levels of noise strength. We quantify the added noise as a percentage strength relative to the initial measurement, ranging from 1% to 99%. We randomly generate noise vectors for each sample at each noise level and add these vectors to the data during training and testing. Specifically, the noise vectors are generated with a standard deviation of

$$\sigma = n \frac{l_{max}}{2} \tag{1}$$

where $n$ is the noise level and $l_{max}$ is the max absolute value of the leadfield for that sample.

During training, noise vectors were randomly generated for each new setup. The results were validated against 50 generated noise vectors (per sample) and results are reported on 100 test vectors (per sample). These noise vectors were saved and stored for consistency across all experiments.

In real EEG measurements, it is common for issues to arise with electrodes. Faulty EEG sensors or loose adhesion to the scalp can cause drastically inaccurate results (e.g., oscillations, drift of over time, amplified measurement noise). For this reason, we also experimented with a different training and testing setup. In addition to the added noise, a small amount measurements would be masked out or removed from the computation. This is done to test each method's prediction consistency when information is lost.

## 3.3 Neural Network Architecture

We propose a transformer architecture for solving the inverse problem. It receives the 76 electrode values from leadfield matrix and the electrode locations as input, and outputs the indices of the predict source region. To illustrate the differences between our approach and other methods, we also provide a baseline fully-connected network that follows the structure and assumptions of other techniques. This network is primarily used to evaluate the effectiveness of the transformer network.

### 3.3.1 Baseline: Multi-Layer Perceptron

As a baseline configuration for comparison purposes, we propose a simple fully-connected network, or Multi-layer Perceptron (MLP), composed of three fully-connected layers, similar to the fully connected setup presented in (Hecker et al., 2022). The hidden layer sizes are 100 and 1000 and each layer is followed by a ReLU nonlinearity. This network assumes fixed input size based on the training data. When values are masked in the input for the dropout experiments, the corresponding channels are simply set to 0.

### 3.3.2 Transformer Network

We propose a transformer network that follows the original structure of the encoder block presented in (Vaswani et al., 2017). Rather than a fixed input equal to the size of the number of channels, each individual channel is treated as its own vector. Each vector is the electrode value appended to its electrode location. This allows the network to learn spatial dependencies across the electrodes, which is crucial in EEG data, where proximity can significantly affect signal interpretation. These four values are then projected into a 512 dimensional space using a linear layer. Similar to Vision Transformers (Dosovitskiy et al., 2021), a learned 512-dimensional embedding token is added to the list of inputs, giving 77 vectors in all. For multi-source prediction, additional learned embedding tokens are added. These inputs are then fed through the standard self-attention mechanisms of the transformer encoder.

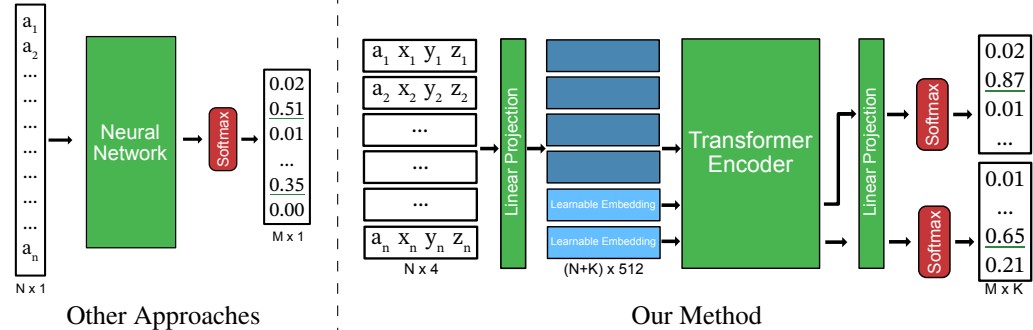

Figure 2: Network diagram for the Transformer model compared to other approaches. In most other methods (Hecker et al., 2021; 2022; Zheng & Guan, 2023), the neural network has a single input vector for the N electrodes and single output predictor vector for the M source regions, with multi source prediction handled by taking the K top values from the output confidences. In comparison, our approach can directly appends location information to the electrode values to treat each as individual token, and can integrate K learned embedding tokens (Dosovitskiy et al., 2021), so that each region source is predicted directly. In our setup, N=76, K=1 for single-source or K=2 for multi-source, and M=1803 for 10mm or M=6790 for 5mm.

After multiple transformer blocks, the values in the embedding token are projected using a linear layer to the number of region clusters. The index with the highest value is considered the source location and the output vector and is put through a cross entropy loss. If multiple sources are present, the closest output to each source is fed through cross entropy loss and each output loss is added together. A visualization of this process in shown in Figure 2.

In our implementation, we use 512 dimensions, 8 attention heads, and 3 transformer layers, with an internal dropout rate of 0.1 to prevent over-fitting. The overall architecture leverages the self-attention mechanism to learn both local dependencies between nearby electrodes and long-range relationships across distant electrodes.

As stated previously, this setup has the benefit of being able to handle any number of input tokens. If data is masked for our drop out experiments, those tokens are simply not processed through the network. Order of the tokens also doesn't matter, removing the risk of misconfiguring the setup for future predictions.

## 4 RESULTS

After training the network, we evaluate our technique on our test set as described in section 3.2 for single-source, multi-source, and dropout tests. Two metrics are reported to evaluate our results: the technique's accuracy in predicting the correct region (Acc.) andthe average Euclidean distance between the predicted region center and the correct region center, this being equivalent to Mean Localization Error (MLE).

### 4.1 SINGLE-SOURCE REGION PREDICTION

We evaluate the ability of the transformer networks to correctly predict the source region. We compare to the baseline MLP network described in Section 3.3.1. The transformer network was trained using a learning rate of 1e-4 using the OneCycle learning rate scheduler (Smith & Topin, 2019) for 250 steps on 10mm and 300 steps on 5mm. The MLP network was trained at 1e-4 for 150 steps and the 1e-5 for an additional 50 steps. We found that both networks converged after these durations.

Additionally, we compare to eLORETA (Pascual-Marqui, 2007) and LCMV beamformer (Van Veen et al., 1997), two classical source localization methods that rely on optimal spatial filter design. As described in Section 3.2, 100 noise vectors for each brain region are saved for testing at each noise level. These noise vectors are added to data before feeding into the classical approaches to allow for fair comparisons.

Table 1: Comparison of the transformer network to the classical eLORETA method, ConvDip approach, and baseline MLP approach on the **10mm** regions. Average Accuracy (Acc.) and Euclidean Distance (MLE) in millimeters are reported for a test set of 100 different noise vectors per possible region. The transformer network performs better on every metric compared to the other techniques.

| Method \Noise | Acc. ↑ | | | | | | MLE ↓ | | | | | |
|---|---|---|---|---|---|---|---|---|---|---|---|---|
| | 1% | 10% | 25% | 50% | 75% | 99% | 1% | 10% | 25% | 50% | 75% | 99% |
| LCMV | 1.000 | 0.999 | 0.965 | 0.594 | 0.345 | 0.238 | 0.000 | 0.002 | 0.517 | 7.924 | 14.29 | 17.79 |
| eLORETA | 1.000 | 0.999 | 0.986 | 0.869 | 0.527 | 0.307 | 0.000 | 0.002 | 0.203 | 4.161 | 14.04 | 25.20 |
| ConvDip | 1.000 | 0.999 | 0.994 | 0.944 | 0.814 | 0.639 | 0.000 | 0.003 | 0.089 | 1.278 | 5.837 | 14.35 |
| MLP | 1.000 | 0.999 | 0.992 | 0.940 | 0.819 | 0.651 | 0.000 | 0.003 | 0.125 | 1.359 | 5.611 | 13.73 |
| **Transformer** | 1.000 | **1.000** | **0.998** | **0.960** | **0.848** | **0.686** | 0.000 | **0.000** | **0.034** | **0.877** | **4.647** | **12.40** |

Table 2: Comparison of the transformer network to other approaches on on the **5mm** regions. The transformer network performs better on both metrics in most cases, especially at high noise levels.

| Method \Noise | Acc. ↑ | | | | | | MLE ↓ | | | | | |
|---|---|---|---|---|---|---|---|---|---|---|---|---|
| | 1% | 10% | 25% | 50% | 75% | 99% | 1% | 10% | 25% | 50% | 75% | 99% |
| LCMV | 1.000 | 0.999 | 0.856 | 0.336 | 0.173 | 0.114 | 0.000 | **0.006** | 1.748 | 11.29 | 16.46 | 19.15 |
| eLORETA | 1.000 | 0.998 | 0.942 | 0.607 | 0.295 | 0.141 | 0.000 | 0.013 | 0.685 | 7.478 | 18.04 | 28.03 |
| ConvDip | 1.000 | 0.999 | 0.983 | 0.851 | 0.623 | 0.424 | 0.000 | 0.008 | 0.212 | 2.841 | 10.07 | 19.74 |
| MLP | 1.000 | 0.998 | 0.983 | 0.858 | 0.645 | 0.450 | 0.000 | 0.013 | **0.209** | 2.674 | 9.319 | 18.57 |
| **Transformer** | 1.000 | **0.999** | **0.986** | **0.878** | **0.669** | **0.471** | 0.000 | 0.055 | 0.341 | **2.246** | **8.602** | **17.79** |

The average results over all brain regions and 100 noise vectors per brain region are provided. The results for the 10mm spacing are presented in Table 1 and the left side of Figure 3. The results for the 5mm spacing are presented in Table 2 and right side of Figure 3. As these results show, the different approaches perform with similar accuracy when there are low levels of noise present. However, MLP and transformer models have significantly higher accuracy at high levels of noise, with the transformer model consistently outperforming the MLP model by 2%-3% accuracy. In addition to predicting source location with higher accuracy, our approach also runs significantly faster in the forward pass than the classical approaches. This illustrates that a well-trained model could provide real-time analysis in future applications.

## 4.2 MULTI-SOURCE REGION PREDICTION

In comparison to traditional techniques, the transformer excels at handling multi-source setups. While most methods assume the top confidence values in the output vector correlate with source locations, the transformer can explicitly handle multi-source prediction by increasing the number of appended learned embedding vectors in the attention mechanism.

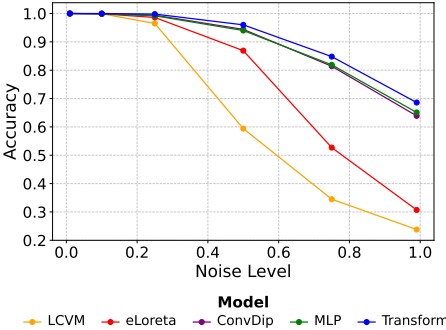 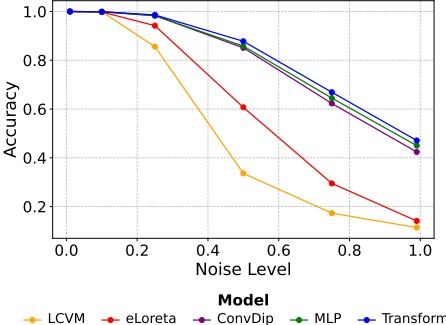

Figure 3: A graph of the accuracy of each method on the 10mm (left) and 5mm (right) regions. The transformer model does better than the classical methods and other deep learned approaches.

Table 3: Comparison of the transformer network to other approaches on on the 10mm regions when predicting two sources of similar strength. While the other methods break down as noise level increases in a multi-source setup, the transformer maintains reasonable performance up to 50% noise.

| Method \ Noise | Acc. ↑ | | | | | | MLE ↓ | | | | | |
|---|---|---|---|---|---|---|---|---|---|---|---|---|
| | 1% | 10% | 25% | 50% | 75% | 99% | 1% | 10% | 25% | 50% | 75% | 99% |
| LCMV | 0.215 | 0.220 | 0.220 | 0.199 | 0.165 | **0.155** | 21.22 | 21.23 | 22.04 | 23.28 | 25.38 | **26.60** |
| eLORETA | 0.019 | 0.011 | 0.012 | 0.001 | 0.005 | 0.004 | 27.69 | 28.44 | 29.04 | 28.98 | 29.27 | 29.75 |
| ConvDip | 0.991 | 0.915 | 0.663 | 0.268 | 0.082 | 0.024 | 0.149 | 0.377 | 6.731 | 21.35 | 33.27 | 41.54 |
| MLP | 0.933 | 0.784 | 0.495 | 0.195 | 0.047 | 0.024 | 1.189 | 3.599 | 10.94 | 23.51 | 34.85 | 43.44 |
| Transformer | **0.995** | **0.986** | **0.863** | **0.564** | **0.297** | 0.138 | **0.128** | **0.203** | **2.100** | **9.955** | **20.57** | 30.87 |

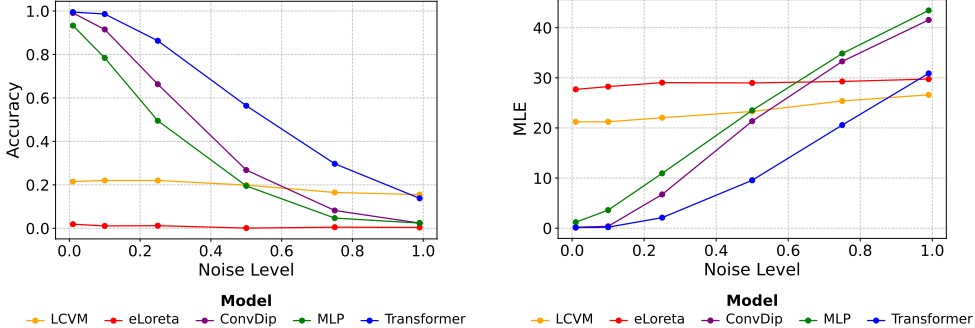

Figure 4: A graph of the accuracy (left) and mean localization error (right) of each method on a two-source setup. The transformer model does better than the classical methods and other deep learned approaches on multi-source prediction.

For our experiments, two sources of similar strength were generated from the forward solution, the distances between the sources being constrained to be 10cm apart. For training and testing, the model must predict the two dipoles locations. The loss function is modified to be permutation-invariant so that no assumptions are made about the order of the predicted dipoles. This is done by exclusively pairing predicted and actual sources in a way that globally minimizes the distances between predicted and actual source locations. This also prevents collapse of the model to predict the same source location twice. The results of this experiment for all the compared methods are presented in Table 3 and visualized in Figure 4.

### 4.3 HANDLING DROPOUT OF ELECTRODES

In the collection of real EEG data, it is very common to have a few electrodes with difficulties in measurements during data collection. Electrodes could be faulty, poorly calibrated, or physically disconnect from the scalp, providing inaccurate measurements. These "bad" electrodes could be clustered around a particular scalp area leading to problems in accurate detection especially if the dipolar-nature of the underlying active brain source is disrupted. Many source localization methods are ill-equipped to handle such scenarios, but our proposed transformer model can naturally handle such situations since it can process any number of inputs at runtime.

To test our transformer model on its ability to generalize in such situations, we simulate these conditions by randomly dropping 5 electrode values from each training and testing sample. This forces the model to learn and make predictions even in the presence of missing data. The results of this experiment are presented in Table 4. As is shown, the transformer model continues to perform with high accuracy and low localization error when data is missing. At 99% noise, the MLE from 12.40 to 14.34 when dropout was added, showcasing the transformers exceptional resilience to missing data when compared to other methods.

Table 4: Comparison of the transformer network to the classical eLORETA method, ConvDip approach, and baseline MLP approach on the 10mm regions when dropping 5 electrodes randomly. The transformer network shows minimal performance drop off when presented with missing information.

| Method \Noise | Acc. ↑ | | | | | | MLE ↓ | | | | | |
|---|---|---|---|---|---|---|---|---|---|---|---|---|
| | 1% | 10% | 25% | 50% | 75% | 99% | 1% | 10% | 25% | 50% | 75% | 99% |
| LCMV | 1.000 | 0.999 | 0.961 | 0.581 | 0.326 | 0.223 | 0.000 | 0.002 | 0.575 | 8.320 | 14.95 | 18.35 |
| eLORETA | 1.000 | 0.999 | 0.982 | 0.796 | 0.495 | 0.282 | 0.000 | 0.003 | 0.255 | 4.837 | 15.44 | 26.91 |
| ConvDip | 1.000 | 0.999 | 0.991 | 0.929 | 0.780 | 0.600 | 0.001 | 0.018 | 0.149 | 1.699 | 7.215 | 16.54 |
| MLP | 0.999 | 0.997 | 0.974 | 0.894 | 0.755 | 0.592 | 0.022 | 0.064 | 0.511 | 2.775 | 8.245 | 16.79 |
| Transformer | 1.000 | **1.000** | **0.996** | **0.949** | **0.820** | **0.649** | 0.001 | **0.001** | **0.059** | **1.190** | **5.814** | **14.34** |

## 4.4 ADDITIONAL ABLATIONS

In addition to the presented transformer model, we experimented with various network configurations and encoding setups. Ultimately, we found the model presented represents an optimal setup. These additional ablations can be found in the appendix.

## 5 DISCUSSION AND CONCLUSIONS

We have presented a transformer-based model for EEG single-source and multi-source localization and have demonstrated its improved performance over baseline approaches, especially in scenarios with higher levels of noise or dropout of electrodes. In single-source tests, the transformer maintained the highest accuracy and lowest mean localization error across the majority of noise levels, outperforming both classical methods and the deep learning baselines. Beyond single-source settings, the transformer extends naturally to multi-source prediction by increasing the number of learned query embeddings and using a permutation-invariant loss. In two-source experiments with sources 10 cm apart, the transformer retained strong performance up to moderate noise (e.g., Acc.=0.564 and MLE=9.955 mm at 50% noise), whereas other methods degraded much more rapidly. The model also showed resilience to realistic data quality issues. In the transformer model, dropping 5 electrodes randomly only modestly decreased performance relative to the no-dropout case. For example, at 99% noise on the 10 mm regions, accuracy only dropped by 3.7% and mean localization error only went up by 1.96 mm. Practically, these results indicate that the proposed transformer model can better tolerate high sensor noise, scale to multiple concurrent sources, and handle missing channels at inference time. These are desirable properties as EEG source localization techniques move towards application to real data in clinical settings with imperfect monitoring and setups.

A current limitation of this work is the reliance on a large amount of synthetically generated training and testing data. Ultimately, to be transferable to clinical applications, the transformer model will need to be validated on evoked and resting EEG from human participants (e.g., motor tasks for known ground truth and empirically derived noise) in order to assess its generalization across head models and preprocessing pipelines. Additional practical considerations include verifying adaptability across electrode layouts and uncertainty estimation in predictions.

In future work, this model could be extended to temporal signals. By including recurrent neural networks or additional attention mechanisms, temporal electrode information could be encoded into feature vectors and fed through a similar transformer model. Integration with popular EEG software toolboxes, such as MNE (Gramfort et al., 2013) could also increase usability with temporal data. Additional future work could explore the capabilities of the transformer across different head models and types, further verifying the transformer's versatility.

Finally, we note that there are still many disparate frameworks and standards for the EEG source localization task. By making the code and data of this work publicly available, we hope to provide accessible benchmarks for others to compare against and enable future developments in EEG source localization techniques in the machine learning community.

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

## A    ABLATION: UNCONSTRAINED LOCATION PREDICTION

While our region clustering approach provides effective results, it is worth exploring whether region clustering is needed at all. We explore a network model which is unconstrained in its location prediction. In this setup, the Transformer network is modified to predict an $x, y, z$ location and $n_x, n_y, n_z$ vector direction to indicate a location of the brain surface. This is done by modifying the last projection layer to output 6 values. For baseline comparison, we also modified the Linear Network in a similar way. The $x, y, z$ vector goes through a tanh function, and the $n_x, n_y, n_z$ goes through an L2 normalization. This overall structure is visualized in Figure 5. This prediction is unconstrained and does not require a known number of regions on the brain surface.

For the loss function, we use mean-square error on the Euclidean distance multiplied by an approximation of the angle between the points on a sphere, representing an approximation of the great-circle distance between the two points.

$$s \approx ||\mathbf{p_2} - \mathbf{p_1}||((1 - (\mathbf{n_1} \cdot \mathbf{n_2}))\frac{\pi - 2}{4} + 1) \tag{2}$$

This linear approximation prevents numerical instabilities that would be present in the exact definition. A deeper discussion of these numerical instabilities and the derivation of this linear approximation are presented in the next section.

The results of the unconstrained location task are shown in full in Table 5. As is seen, the transformer model outperforms the MLP model, but both under perform compared to the standard approach and even classical methods. Both the MLP and Transformer models perform reasonably at low noise levels, with accuracies nearing 100%. However, each model underperforms at higher noise levels, with 50% noise accuracies of 0.548 and 0.771 respectively. This shows that the unconstrained approach can learn on electrode data, but may need additional modifications and improvements to be viable at high noise levels. Future work could improve such an approach to allow generalization of prediction without requiring a known number of brain regions.

## B    DERIVATION OF UNCONSTRAINED LOCATION LOSS FUNCTION

In the unconstrained location prediction task presented in Equation 2 is presented as a linear approximation of great-circle distance. The derivation of this approximation is presented here.

First, a sphere can be perfectly defined given two points on the surface and the normals to the surface at those points. The distance of those two points along the surface can be defined using the arc length between those two points. Mathematically,

$$s = r\theta \tag{3}$$

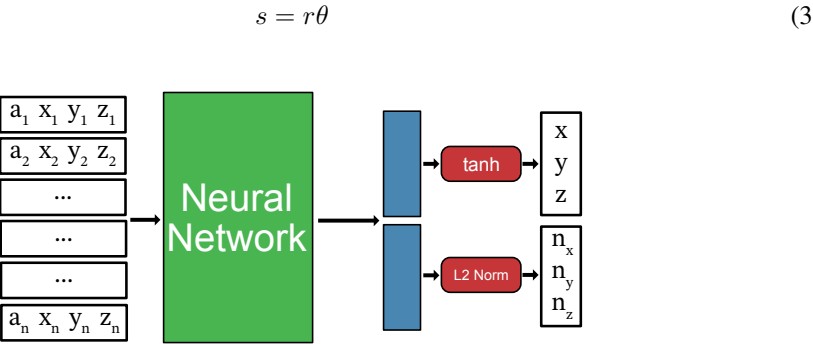

Figure 5: Network configuration for unconstrained location prediction. Rather than generating a vector of prediction confidences per region, six predictions are outputted. Three of those predictions are fed through a tanh function to map to a location within the normalized brain boundaries (-1,1) and the other three are fed through an L2 normalization to generate a unit vector representing the surface normal at that location.

Table 5: Comparison of unconstrained location prediction methods on the 10mm regions. While neither method outperformed the region-prediction based approaches, the transformer network showed significantly better performance over a standard MLP. Future developments of this approach could lead to better generalization across brain models.

| Noise Level | MLP | | Transformer | |
|---|---|---|---|---|
| | Acc. ↑ | MLE ↓ | Acc. ↑ | MLE ↓ |
| 0.01 | 1.000 | 0.517 | 1.000 | 0.472 |
| 0.10 | 0.999 | 0.683 | 0.999 | 0.805 |
| 0.25 | 0.951 | 1.716 | 0.988 | 1.344 |
| 0.50 | 0.548 | 5.378 | 0.771 | 3.730 |
| 0.75 | 0.213 | 10.45 | 0.374 | 8.507 |
| 0.99 | 0.085 | 15.81 | 0.162 | 14.18 |

where $s$ is the arc length, $r$ is the radius of the sphere, and $\theta$ is the angle between the two points. The radius of the sphere can be found by using the Law of Cosines. Specifically,

$$r = \frac{d}{\sqrt{2 - 2\cos\theta}} \tag{4}$$

where $d$ is the Euclidean distance of the two points on the surface. The angle between the two points on the surface matches can be found using the normals of the two points via a simple dot product. Mathematically,

$$\theta = \cos^{-1}(\mathbf{n}_1 \cdot \mathbf{n}_2) \tag{5}$$

where $\mathbf{n}_1$ and $\mathbf{n}_2$ are the unit normals of the points on the surface. Rewriting Equation 3 with these definitions, the equation becomes

$$s = d\frac{\cos^{-1}(\mathbf{n}_1 \cdot \mathbf{n}_2)}{\sqrt{2 - 2(\mathbf{n}_1 \cdot \mathbf{n}_2)}} \tag{6}$$

Note that $d$ can simply be calculated as $||\mathbf{p_2} - \mathbf{p_1}||$.

The right side of Equation 6 provides a numerical instability. Both the inverse cosine and the square-root denominator tend towards 0 as the angle between the two points goes to zero. Even with this division, the range of this function is 1 to $\frac{\pi}{2}$ as the dot product of the normals ranges from 1 to -1. However, the cosine inverse and near-zeros divisions create a numerically instable loss function for purposes of training. Thus, we create a linear function that matches this range and domain. Specifically,

$$f(x) = (1 - x)\frac{\pi - 2}{4} + 1 \tag{7}$$

is a linear function has range of 1 to $\frac{\pi}{2}$ as $x$ goes from 1 to -1. Replacing $x$ with the dot product of the normals and replacing the right side of Equation 6 provides the final approximation as presented previously in Equation 2. Rewritten here,

$$s \approx ||\mathbf{p_2} - \mathbf{p_1}||((1 - (\mathbf{n}_1 \cdot \mathbf{n}_2))\frac{\pi - 2}{4} + 1) \tag{8}$$

This function provides a reasonable approximation for great-circle distance while avoiding the numerical instabilities during training.

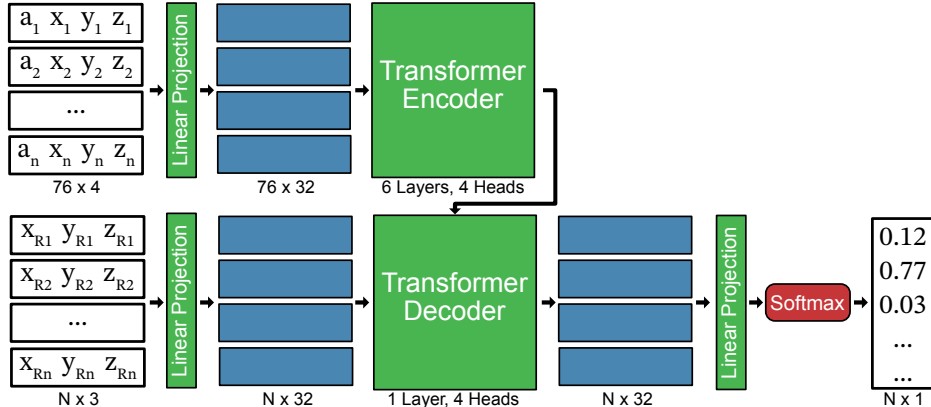

Figure 6: Encoder-Decoder Transformer Structure. In this configuration, the region locations are also projected into the same dimensional space as the EEG measurements and locations. The EEG vectors perform cross attention to the region vectors. Each region vector is then fed through a linear projection and is scored. Due to the large number of regions the dimensionality is reduced to 32 to improve training speed, but ultimately this configuration failed to converge in most training scenarios.

## C   ABLATION: ENCODER-DECODER TRANSFORMER ARCHITECTURE

We experiment with an additional transformer network that uses an encoder-decoder architecture. In our encoder-only architecture, the number of output regions must be kknown at training time since it relies on a linear projection for the classification. If an encoder-decoder architecture is used, the network could cross attend to each region individually to determine the most likely source location. We experiment with such a possibility.

As in the encoder-only architecture, the 76 electrode inputs are appended to their location and projected to 32 dimensional vectors and considered the source tokens. Similarly, the brain region centers are projected into 32 dimensional space and considered the target tokens. This small dimensional size is chosen to account for the large amount of cross attention that will occur. The electrode vectors are fed through a transformer encoder which consists of 6 layers and 4 heads. The output from this network is sent to the decoder and cross-attends to the region vectors. The decoder consists of 1 layer and 4 heads. At the end of the network, each region vector is scored by feeding it through a linear layer and projected to a single value. The vector with the highest value is considered the source location and the lists of values is put through a cross entropy loss. A full illustration of the network architecture is presented in Figure 6.

For this full transformer network, even with the smaller dimensionality space, we find that the network does not generally converge to a solution. If it does, it does not do so quickly or as well as the encoder-only network. Because of the large number of clusters to cross-attend to, the overall accuracy remains very low, peaking at 78% even when just 1% noise is present. It is possible that with a pretraining scheme, large dimensional vectors, and more compute power and time, such a network could be feasible in future work.

## D   ABLATION: ITERATIVE TRAINING APPROACH

The results presented in Tables 1 and 2 required training an individual model exclusively at a single noise level. Multiple models are trained from scratch, even though they are learning the same prediction task, just at various levels of difficulty. It is worth investigating whether a model learned at one noise level can transfer knowledge to a model trained for another level.

In this ablation, we experimented with using an iterative training approach, where the weights from a lower noise level are the starting point for a model trained at a higher noise level. Our motivation for adopting this approach is to improve efficiency by eliminating the need to train separate models for each noise level, which in turn substantially reduces computational costs.

Table 6: Comparison of average accuracy of iterative models on each noise level on the 5mm regions. Both the iterative MLP and Transformer network were trained for 50 epochs per noise level using 1e-4 oneCycle learning rate schedule. The iterative approach does not outperform the standard approach, but does train for all noise levels in a much shorter time.

| Noise Level | Standard MLP | Standard Transformer | Iterative MLP | Iterative Transformer |
|---|---|---|---|---|
| 0.01 | 1.000 | 1.000 | 1.000 | 1.000 |
| 0.10 | 0.998 | 0.999 | 0.998 | 0.999 |
| 0.25 | 0.983 | 0.986 | 0.954 | 0.984 |
| 0.50 | 0.858 | 0.878 | 0.800 | 0.865 |
| 0.75 | 0.645 | 0.669 | 0.593 | 0.653 |
| 0.99 | 0.450 | 0.471 | 0.417 | 0.458 |

Table 7: Comparison of the average accuracy of the transformer model with and without and positional encoding for the 10mm region prediction task. The model with positional encoding performed worse at each noise level. The improved performance of the network without positional encoding is likely due to the EEG measurements being sparse and locations being directly incorporated into the linear projection.

| Noise Level | No Pos. Enc. Acc. ↑ | With Pos. Enc. Acc. ↑ |
|---|---|---|
| 0.01 | 1.000 | 1.000 |
| 0.10 | 1.000 | 0.987 |
| 0.25 | 0.998 | 0.981 |
| 0.50 | 0.960 | 0.879 |
| 0.75 | 0.848 | 0.810 |
| 0.99 | 0.686 | 0.175 |

Our iterative training approach involves training models progressively across increasing noise levels (from 0.01 to 0.99), with each noise level trained for 50 epochs before moving to the next. The model weights are saved after each stage to ensure smoother transitions and stable learning. Both the MLP and Transformer model were evaluated at the currently trained noise level during the iterative process. These results are provided in Table 6.

The iteratively-trained transformer outperformed both the standard-training and iterative-training MLP, particularly at high noise levels. However, accuracy at high noise levels did not surpass standard training for the transformer model at higher epochs (e.g., 250 epochs at 99% noise). The limitation of this approach is that increasing the number of epochs per noise level beyond 50 makes the iterative approach more computationally expensive than single-noise-level training. Thus, the standard training pipeline provides the best results, but the iterative approach could be useful for quickly training effective models at all noise levels.

# E    ABLATION: NETWORK ARCHITECTURE

To verify our selection of neural network architecture, we conduct an ablation study comparing results of various network and hyper-parameter configurations.

## E.1    POSITIONAL ENCODINGS

In most transformer-based architectures for computer vision or natural language processing, a positional encoding is added to outputs of the projection layers to indicate some spatial or sequential location of the input. Thus, we experimented with adding a positional encoding for the electrode locations rather than appending them to the inputs. The results for that experiment are provided in Table 7. As can be seen, an added positional encoding under-performs compared to feeding the positions through the linear layer with the electrode values. The sparse nature of the input may be a factor in eliminating the need for a positional encoding.

Table 8: Ablation study on the learning rate for the 10mm regions. The accuracy is reported.

| | MLP | | | Transformer | | |
|---|---|---|---|---|---|---|
| Noise Level | 1e-5 | 1e-4 | 1e-3 | 1e-5 | 1e-4 | 1e-3 |
| 0.01 | 1.000 | 1.000 | 1.000 | 1.000 | 1.000 | 0.999 |
| 0.10 | 1.000 | 0.999 | 1.000 | 1.000 | 1.000 | 0.996 |
| 0.25 | 0.986 | 0.992 | 0.996 | 0.997 | 0.998 | 0.980 |
| 0.50 | 0.860 | 0.940 | 0.950 | 0.956 | 0.960 | 0.902 |
| 0.75 | 0.617 | 0.819 | 0.827 | 0.836 | 0.848 | 0.745 |
| 0.99 | 0.400 | 0.651 | 0.661 | 0.672 | 0.686 | 0.579 |

## E.2    HYPERPARAMETERS

A hyper parameter study is conducted and the results are presented in the following tables. Table 6 presents the effect of different learning rates on the MLP and Transformer model performance. Table 7 evaluates the effect of the step and one cycle learning rate scheudulers on performance of the models. Tables 8 and 9 does an ablation on the transformer dimensions and layer counts.

These ablations demonstrate that the model parameters and training configurations that are presented in the results of the paper generally provide the best results. However, it is worth noting two exceptions. First, the combination of a 1e-3 learning rate and the OneCycle scheduler on the 10mm cluster for the MLP model outperformed the base MLP model using the step scheduler, but it was found that a similar configuration for the 5mm MLP performed worse than the step scheduler, indicating the MLP's sensitivity to the training configuration. Second, the transformer performed slightly better when increasing the size of the dimensions and the number of layers, but doing so causes a dramatic increase in the needed computational resources and training time.

Table 9: Ablation study on the learning rate scheduler for the 10mm regions. The accuracy is reported.

| | MLP | | | Transformer | |
|---|---|---|---|---|---|
| Noise Level | Step | Onecycle (1e-3) | Onecycle (1e-4) | Step | Onecycle |
| 0.01 | 1.000 | 1.000 | 1.000 | 1.000 | 1.000 |
| 0.10 | 0.999 | 1.000 | 1.000 | 1.000 | 1.000 |
| 0.25 | 0.992 | 0.996 | 0.991 | 0.997 | 0.998 |
| 0.50 | 0.940 | 0.952 | 0.924 | 0.958 | 0.960 |
| 0.75 | 0.819 | 0.836 | 0.797 | 0.847 | 0.848 |
| 0.99 | 0.651 | 0.675 | 0.633 | 0.686 | 0.686 |

Table 10: The accuracy of the transformer model on 10mm regions with various sizes for the projection dimension. While the 768 dimensional transformer performs slightly better than the presented 512 method, it requires significantly more computational resources.

| Noise Level | 64 Transformer | 512 Transformer | 768 Transformer |
|---|---|---|---|
| 0.50 | 0.958 | 0.960 | 0.960 |
| 0.75 | 0.839 | 0.848 | 0.850 |
| 0.99 | 0.676 | 0.686 | 0.690 |

Table 11: An ablation study on the accuracy of the transformer model on 10mm regions when varying number of layers. While the 4 dimensional transformer performs slightly better than the presented 3 layer method, it requires significantly more computational resources and training time.

| Noise Level | 2 Layer Transformer | 3 Layer Transformer | 4 Layer Transformer |
|---|---|---|---|
| 0.50 | 0.957 | 0.960 | 0.962 |
| 0.75 | 0.839 | 0.848 | 0.850 |
| 0.99 | 0.675 | 0.686 | 0.691 |

