# OpenReview forum: "Transformer Networks Enable Robust Generalization of Source Localization for EEG Measurements"
_ICLR.cc/2026/Conference — ICLR 2026 Conference Withdrawn Submission_

### Official Review · Reviewer_UjSU · 2025-10-24

**Soundness:** 1
**Presentation:** 1
**Contribution:** 1
**Rating:** 2
**Confidence:** 4

**Summary:**

This paper addresses EEG source localization using a Transformer-based approach to learn embeddings that separate single-source from multi-source activations. While source localization is a fundamental problem in EEG science, it is traditionally treated as a signal-processing / inverse problem rather than a standard ML/DL task. The authors evaluate their method on simulation data and report promising results. However, the manuscript currently reads more like a biomedical engineering report than a machine-learning conference contribution: crucial methodological details (problem formulation, mathematical derivations, network architecture specifics) are missing, figures are unclear, and experimental choices (e.g., use of simulated-only data) are insufficiently justified. In its present form, the work lacks the rigor and reproducibility expected at ICLR.

**Strengths:**

Tackles an important and nontrivial application: EEG source localization is a core problem with high scientific and translational value.

Bringing contemporary sequence/attention models to this domain is an interesting direction that could spark cross-disciplinary dialogue.

The idea of learning token embeddings to disambiguate single-source vs. multi-source localization is promising and, if convincingly developed, could complement classical inverse methods.

The paper is generally readable at a high level; experiments on simulated data are a reasonable first step to show feasibility.

**Weaknesses:**

1. The manuscript lacks a clear, mathematical statement of the inverse problem being solved. For instance, the mapping from sensor measurements to source distributions, the forward model (lead field), and the exact linear system referenced on page 4, line 210, are not written out or derived. The description of “how the Transformer solves the linear equation” is hand-wavy and insufficient for replication or theoretical assessment.

2. Figure 2 is unclear, and the block diagram deviates from standard DL schematic conventions. Individual blocks are not labeled with layer types, shapes, activation functions, or parameter counts. Treating network components as black boxes is unacceptable at ICLR. The community requires a full specification (or code) and ideally, ablation studies demonstrating the contribution of each module.

3. The experiments are based entirely on simulated EEG; however, simulation design critically determines the difficulty of localization tasks (SNR, source depth, number of sources, correlated sources, head model mismatch, noise characteristics). The paper does not sufficiently justify simulation choices nor discuss the gap to real EEG, which weakens claims about practical relevance.

4. Classical and widely used baselines in the EEG source localization literature (eLORETA, sLORETA, L2-norm inversion, LCMV beamformer, etc.) are not described in detail nor thoroughly compared. Even if these are not DL methods, they are essential reference points; omission undermines empirical conclusions.

5. Related work conflates machine learning and deep learning approaches and lacks a clear taxonomy. This mixing reduces clarity and makes it difficult to position the contribution relative to existing work.

6. There is no systematic exploration of how architectural choices, tokenization strategies, or training regimes affect localization performance, nor of robustness to common real-world problems (e.g., mis-specified lead field, nonstationary noise, inter-subject variability).

**Questions:**

1. Please provide a formal problem statement. Write the forward model (lead field), the inverse problem formula, and show explicitly which linear system you refer to on page 4, line 210. How are sensors, sources, and noise modeled mathematically?

2. Detail the network architecture completely. For every block in Fig.2, specify layer types, input/output shapes, activation functions, normalization layers, dropouts, and parameter counts. If code is provided, indicate the exact commit/hash for reproducibility.

3. Explain the Transformer mapping to the linear inverse problem. Mechanistically, how does the attention mechanism solve or approximate the linear inversion? Is the Transformer acting as a learned regularizer, an approximate solver, or a classifier of source counts? Provide equations or pseudo-code.

4. Provide full descriptions and implementations of baselines. Include eLORETA, sLORETA, L2, LCMV beamformer, and any other standard methods. Describe hyperparameters and how they were tuned. Compare against them quantitatively under identical simulation conditions.

5. Justify the simulation protocol and add realism. Describe simulation parameters (SNR range, depth distribution, number of sources, correlated sources, head model robustness). Consider adding experiments that test robustness to model mismatch and show performance on at least one real EEG dataset.

6. Add ablation and sensitivity analyses. Ablate major components (tokenization, attention vs. no-attention, depth of Transformer, positional encodings). Report how performance changes with SNR and the number of simultaneous sources.

7. Clarify evaluation metrics and statistical reliability. State how many simulation trials were run and whether cross-validation was used.

8. Rework the Related Work section. Separate classical source localization (signal-processing inverse methods), machine learning approaches, and deep learning methods. Discuss the most relevant prior art and clearly state novelty.

---

### Official Review · Reviewer_Z5g5 · 2025-10-26

**Soundness:** 2
**Presentation:** 2
**Contribution:** 1
**Rating:** 0
**Confidence:** 5

**Summary:**

The paper proposes a transformer that ingests EEG data with 3D electrode coordinates and predicts one or two source locations. Training is entirely synthetic using a single FEM head model with 76 electrodes. Evaluation spans multiple noise levels, two spatial discretization (10 mm and 5 mm), and a mild channel-dropout setting (removing five electrodes). Across these settings, the method outperforms eLORETA, LCMV, and two deep-learning baselines, with improvements on the order of a few millimeters.

**Strengths:**

The model beats the benchmarks by a few millimeters.

**Weaknesses:**

Despite the claim of “ROBUST GENERALIZATION,” the evidence points to limited generalization:

1. Same head model for train/test, causing “inverse crime”.
2. Per-noise models. Tables 1–2 rely on training a separate model for each noise level. In practice, the noise level is unknown, so this setup is not deployable on real data.
3. Narrow multi-source evaluation. Only two sources of similar strength, fixed 10 cm apart, are tested. There is no sweep over spacing, correlation, or strength imbalance, limiting the generality of the multi-source claims.
4. K must be known a priori. The approach requires specifying the number of sources in advance to select the appropriate model, which is unrealistic in real-world scenarios. This reflects design-level limitations, not the issue that additional validation alone can remedy.
5. The improvement of LE over other DL method is only about 1mm, which has limited practical impact in real applications.

**Questions:**

1. What is the performance of using different head model during training and testing? (5 mm for training, 10 mm for testing)
2. How did you calculate Acc and LE for distributed imaging methods? Are you taking the maximum?

---

### Official Review · Reviewer_JKb3 · 2025-10-29

**Soundness:** 3
**Presentation:** 2
**Contribution:** 2
**Rating:** 2
**Confidence:** 4

**Summary:**

The paper introduces a new transformer based deep learning architecture for the estimation of the underlying sources from scalp EEG signals essentially solving an ill-posed problem using a neural network modeling procedure. The network is trained on extensively simulated data using the Ernie head model from the SimNIBS simulation toolbox and evaluated in its ability to reconstruct the simulated sources based on the forward models generated EEG signals. The approach is compared some prominent source localization procedures and found to improve upon the simulated source estimates especially in noisy setting and in the presence of more than one source (i.e., two sources being active).

**Strengths:**

•	The methodology is sound and appears useful and noise robust.

•	The approach outperforms existing methodologies compared against.

•	The paper and developed methodology is easy to follow.

Originality: The novelty of the paper is somewhat limited in light of architectures and transformer designs of recent EEG foundation models that the paper does not relate to. The contribution is mainly the considered use of transformer architectures for EEG inverse problems and use of extensive simulated data to learn a deep learning framework to perform the source reconstruction.


Quality: The paper is well written and the experimentation using simulated data sound.

Clarity: The paper is clear and easy to follow and the methodology generally well presented, but the paper is in need of some careful proof reading.

Significance: I find the significance of the contribution limited. In particular, the developed methodology is not very novel whereas the experimentation only considers synthetic data making the impact of the methodology in practice very unclear. Considering some real data where the source activation is known (i.e. visual paradigms-> visual cortex activation, face perception - fusiform face area activation, etc. ) and compared to existing source localization on real data would substantially help position the relevance and impact of the paper.

**Weaknesses:**

•	The major weakness is that the whole procedure relies on simulated data and it is therefore very unclear how practical useful the developed methodology is. The approach could easily have been trained on extensive simulations and the learned backward mapping applied to real EEG data with known source activations such as face perception (Fusiform face area activation) etc. By failing to include any real world experimentation the impact of the developed method is very unclear.

•	The use of transformers for positional encoding and channel agnostic training etc. is not new and the paper needs to relate their approach to the by now many foundation modeling approaches for EEG data that are similarly channel agnostic in the training and using channel information in the embeddings and transformer based architecture (see also questions).

•	The results do not include any error bars and therefore the statistical significance is difficult to assess.

•	The paper can be improved in its readability by careful proof-reading, see also minor comments below.

Minor comments on proof reading:

difficult generalize the -> difficult to generalize the

large amounts simulated data -> large amounts of simulated data

, and have been explored -> have been explored

heavily evaluated -> extensively evaluated

We empirically found the that the presented approach -> We empirically found that the presented approach

Third research question is poorly formulated please revise this or at least remove “through” in the following sentence: “We conduct through evaluations and comparisons, showing our model outperforms other architectures and methods on single- and multi-source prediction for high-noise and drop out scenarios.”

is only be determined -> can only be determined

Figure 1 caption please check (left) and (right) in the caption. Seems these left and right references to the images are incorrect.

drift of over time -> drift over time

amount measurements -> amount of measurements

of the predict source -> of the predicted source

andthe -> and the

to Mean Localization Error -> to the Mean Localization Error

**Questions:**

How do the presented channel agnostic encoding differ from recent channel agnostic EEG foundation models such as the following procedures, see also:

Yang, Chaoqi, M. Westover, and Jimeng Sun. "Biot: Biosignal transformer for cross-data learning in the wild." Advances in Neural Information Processing Systems 36 (2023): 78240-78260.

Jiang, Wei-Bang, Li-Ming Zhao, and Bao-Liang Lu. "Large brain model for learning generic representations with tremendous EEG data in BCI." arXiv preprint arXiv:2405.18765 (2024).

Wang, Jiquan, et al. "Cbramod: A criss-cross brain foundation model for eeg decoding." arXiv preprint arXiv:2412.07236 (2024).

Zhou, Yuchen, et al. "CSBrain: A Cross-scale Spatiotemporal Brain Foundation Model for EEG Decoding." arXiv preprint arXiv:2506.23075 (2025).

Why is the procedure not applied to real EEG data with task activations in which the sources are know (i.e. Face Activation -> Fusiform face area, visual stimuli -> occipital lobe/V1 activity etc.) This is standard for evaluating source localization procedures which this paper aims to contribute towards. Without such evaluation the paper does not have clear impact assessments – why is this important evaluation step omitted here?

What are the uncertainty of the estimates provided in the Tables – why are error bars not here reported?

---

### Official Review · Reviewer_kPUA · 2025-10-29

**Soundness:** 2
**Presentation:** 1
**Contribution:** 2
**Rating:** 2
**Confidence:** 3

**Summary:**

This paper proposes a transformer architecture for source reconstruction from sensor measurements. A transformer is well suited here as it can handle a variable number of inputs and outputs, which is important for EEG as there can be a variable number of sensors and different number of desired outputs. The paper shows that this architecture works better than some classical source reconstruction algorithms like eLORETA and other learned alternatives.

**Strengths:**

### Originality/significance
- This paper studies an important problem in EEG research broadly and in using deep learning applied to it in particular.
- Handling a different number of inputs is really important here and not as common in more typical deep learning setups.
- It’s always good to ablate design choices in the community, like different model architectures.


### Clarity
- The paper’s contributions and main claims are clearly laid out.
- Although fairly involved on the neuroscience side for a typical ICLR paper the authors try to make it accessible with a decently thorough related work section.


### Technical soundness
- The paper presents lots of ablations and different settings to see how well the proposed model works in various relevant setups.
- Decent baseline comparisons are given.

**Weaknesses:**

### Clarity
- Does the paper present its titular claim? Specifically, that these networks generalize, and if so then how so? Many important details in the development, training, and test setups are omitted so this is hard to infer. No mention of generalization is given in the contributions or discussion, at least not explicitly. It is really important to discuss which kind of generalization the authors are referring to - for example, based on what I understood from the experiment setup, the train and test data are in all cases from the same distributions. So is this in distribution generalization, when training and testing on the same subject? Depending on the exact settings this would beg other relevant questions, like whether this is the most important kind of generalization for EEG data vs something like cross machine or cross subject generalization, which would allow harmonization, and how it compares to (presumably) robust approaches like eLORETA in those cases.
- As mentioned, many very important technical details are missing - what’s the train/test split? How were hyperparameters chosen? How many parameters do the networks have? It’s hard to gauge the result’s reliability, how expensive these experiments are, etc. without knowing this.
- In the abstract it’s mentioned that “These approaches, however, make underlying assumptions that make it difficult to generalize the results past their original training setups” - it would be good to expand on this, it seems important but isn’t thoroughly (or at least sufficiently explicitly) discussed. I presume the authors mean different machines having different numbers of electrodes, faulty ones, or something similar but this should be said.
- Minor - there are some typos and unclear phrasing throughout the paper, e.g. “and have been explored” (line 46), “transformer architectures that do exist” (line 49).
- Minor - define MLE before using it (line 62)
- The region spacing/clustering defined in lines 69-72 is pretty unclear.
- Minor - worth explicitly stating that this setup requires MRI scans/structurals to work, not just the EEG data, which technically could be fit to a template.
- Minor - would be good to explicitly say why source localization is ill posed, for the new reader.
- Minor - in lines 121-127 it would be nice to give a source for these problems of “classical” source reconstruction methods, and later follow up on which of these this learned approach solves.
- Minor - LCMV should be better explained.
- Minor - some of lines 164-172 can go in the background section.
- “There are specific considerations in configuring a machine learning model to solve the inverse problem. We explore those considerations” - this should be better emphasized throughout as it’s quite important. Currently the next few sections in the paper have a lot of “what” and could use more “why”, explaining the rationale between these design choices, which problems they solve, etc. This is especially true for lines 205-220.
- Lines 315-320 - is this a single or two different models’ setups? I’m confused.
- Line 323 - don’t all methods see the same input? It would be good to make it clearer how these comparisons are fair, that’s important.
- How big are the datasets?
- Region clustering should be more clearly introduced.
- Minor - a plot in Appendix B would be nice, more clearly show the approximation that’s being used.


### Originality/significance
- Isn’t source reconstruction much more prevalent for MEG than EEG [1]? Why focus specifically on EEG then? There might be some applications where it’s important for EEG and MEG isn’t suitable, if these are motivations then it’s good to discuss them. Regardless, discussing the EEG specific motivation would be good, assuming it’s a convincing one.
- Follow up, more of a question than a weakness - would this be relevant for MEG? Why/why not? Even a speculative discussion would be interesting.
- Is the main contribution here the general setup or specifically the use of transformers? Although the contributions section emphasizes the transformers, how do we know the general training setup isn’t what’s good here? I don’t believe the ablations rule this out. This is important as given many moving parts what makes it work can be made muddy, see the semi recent HRM paper [3] and the ensuing discussion [4].


### Technical soundness
- Assuming I properly understood the data generating process and train/test splits are in distribution, to what extent can the model be said to generalize? It’s important to be precise about the kind of generalization asserted and explicitly mention for which applications it’s important and for which it isn’t.
- Are the model comparisons fair? Without details such as the number of parameters it’s hard to say if the transformer performed best due to having a better inductive bias, more/fewer parameters, or something else entirely. Many such details should be accounted for and made clear they’re accounted for.
- Why simulated data? It can be used for some simple experiments and proof of concepts, for sure, but at the end of the day real world performance is what counts. Datasets like [2] have been used for machine learning EEG source reconstruction as far as I’m aware, so this doesn’t seem to be a barrier.
- The ablations are presented throughout (e.g. lines 82-85) as being something done after development, where coincidentally the model is already basically optimal. This feels weird, I recommend changing how you discuss them to better elucidate the different design choices, see [5,6] for examples of nice ablations and how they’re related to the design choices, architecture, etc.
- Regarding the first contributions point on variable number of sensors, this is only slightly demonstrated, and based on what I understood not in the form of generalization (training on N sensors and testing on M) - it would be good to make this much clearer, e.g. training on N and testing on 2N or N/2.
- In lines 290-295, what do you mean by highest index - the last token? If not then is this selection a differentiable process, or do you use a straight through estimator?
- Were the hyperparameters for the ConvDip baseline tuned? If it wasn’t, I assume that’s why the MLP performed better.
- Isn’t the setting described in lines 407-415 very artificial? This can be fine to demonstrate something or test a method but wouldn’t the transformer then perform better here as it has some biases the MLP/CNN don’t?
- When saying a model is optimal it is always optimal only within some constraints - for example, more data would probably help. It’s important to specify these constraints (“the budget”) when discussing a model’s optimality, and how this may change under different settings.
- Why not always use different noise? Given the same set of noise vectors couldn’t the networks potentially be memorizing noise patterns?

**Questions:**

- (lines 40-41) is the slowness of EEG an issue in practice? I thought most source reconstruction based analysis are done offline, but perhaps I’m unaware of some real time clinical use there may be.
- Especially given the use of an ML model for source reconstruction, how well does the model do with “sharp” highly localized outputs? I’d imagine it might have some smearing, especially in the underdetermined case, although same goes to some extent for other methods - would be cool to compare.
- Did you consider using something like an INR based approach for modelling the reconstruction? That totally circumvents segmentation issues and can include an implicit forward model.
- Any ideas why previous works didn’t try using transformers? E.g. is it due to data hungriness/scarcity? Would be nice to even just speculate (lines 151-153)
- How does the data generation setup in section 3.2 compare to previous papers you cited?
- Is the noise used in section 3.2 typical, e.g. where it’s not localized?
- Why focus on surface head regions and not volumes?
- What are realistic noise levels? Is it not possible, and in some cases even likely, to have >100% noise? If high noise is prevalent, based on table 3 wouldn’t the classical methods be better?

## Conclusion

The paper gets at an interesting, relevant problem, and fundamentally takes a potentially promising approach, but lacks many technical details and doesn’t sufficiently demonstrate its main point of generalization. Sadly at its current state I don’t think the paper’s ready for ICLR.

### References
- [1] Puce, Aina, and Matti S. Hämäläinen. "A review of issues related to data acquisition and analysis in EEG/MEG studies." Brain sciences 7.6 (2017): 58.
- [2] Wakeman, Daniel G., and Richard N. Henson. "A multi-subject, multi-modal human neuroimaging dataset." Scientific data 2.1 (2015): 1-10.
- [3] Wang, Guan, et al. "Hierarchical Reasoning Model." arXiv preprint arXiv:2506.21734 (2025).
- [4] https://arcprize.org/blog/hrm-analysis
- [5] Jolicoeur-Martineau, Alexia. "Less is More: Recursive Reasoning with Tiny Networks." arXiv preprint arXiv:2510.04871 (2025).
- [6] He, Kaiming, et al. "Masked autoencoders are scalable vision learners." Proceedings of the IEEE/CVF conference on computer vision and pattern recognition. 2022.

---

### Official Review · Reviewer_zWjn · 2025-11-01

**Soundness:** 2
**Presentation:** 2
**Contribution:** 2
**Rating:** 2
**Confidence:** 5

**Summary:**

This paper applies transformer networks to the EEG source localization problem. Each electrode’s voltage and spatial coordinates are treated as tokens, and learnable query embeddings are used to predict cortical source regions. The authors train entirely on synthetic EEG generated from a finite element head model with different noise levels and electrode dropout, claiming robust generalization over classical solvers and prior deep networks.

**Strengths:**

1.	The general idea of exploring transformers for EEG inverse modeling is interesting and timely. EEG source localization is a long-standing ill-posed problem, and using attention to model spatial relations among electrodes could in principle offer flexibility and robustness.
2.	The implementation is technically consistent and clearly described. The synthetic pipeline is based on a realistic finite element (Ernie) head model and includes noise and electrode dropout, which at least provides a controlled evaluation environment.

**Weaknesses:**

1.	The study is limited entirely to synthetic data, with no validation on real EEG recordings or cross-head simulations. The synthetic setup is overly simplified, relying only on single- and dual-dipole sources without any functionally realistic or temporally correlated activity. This disconnect makes the results biologically uninformative and raises major doubts about external validity.
2.	The model design rests on strong and unexamined assumptions. The use of a fixed number K of learnable query embeddings assumes that brain sources are a finite set of discrete, independent generators—a restrictive prior that is neither justified theoretically nor analyzed empirically. Furthermore, the method ignores EEG’s inherent temporal structure. A spatial-only transformer using static electrode values fails to exploit the temporal resolution that defines EEG, undermining the supposed benefit of attention modeling.
3.	The experimental validation is limited and somewhat outdated. The chosen baselines (eLORETA, LCMV, ConvDip, MLP) are older and do not fully reflect recent advances such as graph-based or FEM-constrained deep models. More importantly, the model’s potential strength—its ability to generalize across head geometries or electrode configurations—is not examined.
4.	The reported metrics have limited relevance to realistic or clinical scenarios. Accuracy and mean localization error on synthetic data do not necessarily reflect whether the method can achieve useful precision in real tasks, such as identifying functional activations or epileptic foci. Without testing under such conditions, the robustness demonstrated here should be interpreted with caution rather than as evidence of practical readiness.

**Questions:**

1.	How would the proposed model perform on real EEG data or cross-subject simulations? Without such validation, it is difficult to assess whether the observed improvements translate beyond synthetic setups.
2.	The method assumes a fixed number of learnable query embeddings K. How sensitive is performance to this assumption, and can the model adapt when the number of active sources is unknown?
3.	Given EEG’s inherently temporal nature, have you considered extending the approach to spatio-temporal transformers, or can you justify why temporal modeling is unnecessary for the presented results?

---

### Note · Authors · 2025-11-20

I have read and agree with the venue's withdrawal policy on behalf of myself and my co-authors.